

# Reliability and concurrent validity of the iPhone® Compass application to measure thoracic rotation range of motion (ROM) in healthy participants

James Furness[1,2], Ben Schram[1,2], Alistair J. Cox[2], Sarah L. Anderson[2] and Justin Keogh[3,4,5]

[1] Water Based Research Unit, Bond Institute of Health and Sport, Bond University, Gold Coast, Queensland, Australia
[2] Department of Physiotherapy, Faculty of Health Science and Medicine, Bond University, Gold Coast, Queensland, Australia
[3] Faculty of Health Science and Medicine, Bond University, Gold Coast, Queensland, Australia
[4] Sports Performance Research Centre New Zealand, AUT University, Auckland, New Zealand
[5] Faculty of Science, Health, Education and Engineering, University of the Sunshine Coast, Sunshine Coast, Australia

Corresponding author
James Furness, jfurness@bond.edu.au

## ABSTRACT

**Background.** Several water-based sports (swimming, surfing and stand up paddle boarding) require adequate thoracic mobility (specifically rotation) in order to perform the appropriate activity requirements. The measurement of thoracic spine rotation is problematic for clinicians due to a lack of convenient and reliable measurement techniques. More recently, smartphones have been used to quantify movement in various joints in the body; however, there appears to be a paucity of research using smartphones to assess thoracic spine movement. Therefore, the aim of this study is to determine the reliability (intra and inter rater) and validity of the iPhone® app (Compass) when assessing thoracic spine rotation ROM in healthy individuals.

**Methods.** A total of thirty participants were recruited for this study. Thoracic spine rotation ROM was measured using both the current clinical gold standard, a universal goniometer (UG) and the Smart Phone Compass app. Intra-rater and inter-rater reliability was determined with a Intraclass Correlation Coefficient (ICC) and associated 95% confidence intervals (CI). Validation of the Compass app in comparison to the UG was measured using Pearson's correlation coefficient and levels of agreement were identified with Bland–Altman plots and 95% limits of agreement.

**Results.** Both the UG and Compass app measurements both had excellent reproducibility for intra-rater (ICC 0.94–0.98) and inter-rater reliability (ICC 0.72–0.89). However, the Compass app measurements had higher intra-rater reliability (ICC $= 0.96-0.98$; 95% CI [0.93–0.99]; vs. ICC $= 0.94-0.98$; 95% CI [0.88–0.99]) and inter-rater reliability (ICC $= 0.87-0.89$; 95% CI [0.74–0.95] vs. ICC $= 0.72-0.82$; 95% CI [0.21–0.94]). A strong and significant correlation was found between the UG and the Compass app, demonstrating good concurrent validity ($r = 0.835$, $p < 0.001$). Levels of agreement between the two devices were 24.8° (LoA −9.5°, +15.3°). The UG was found to consistently measure higher values than the compass app (mean difference 2.8°, $P < 0.001$).

**Conclusion**. This study reveals that the iPhone® app (Compass) is a reliable tool for measuring thoracic spine rotation which produces greater reproducibility of measurements both within and between raters than a UG. As a significant positive correlation exists between the Compass app and UG, this supports the use of either device in clinical practice as a reliable and valid tool to measure thoracic rotation. Considering the levels of agreement are clinically unacceptable, the devices should not be used interchangeably for initial and follow up measurements.

## INTRODUCTION

Thoracic spine mobility is a key component in many water-based sports including surfing, swimming and stand up paddleboarding (SUP) (*Furness, 2015*; *Pollard & Fernandez, 2004*; *Schram, 2015*). In surfing, thoracic rotation is a critical movement during wave riding as it allows the surfer to produce sufficient torque to turn and perform maneuvers (*Furness et al., 2016*). It is also hypothesised that a reduction in thoracic extension during the paddling phase of surfing could potentially result in greater stress on the lumbar or cervical spine (*Furness et al., 2014*). Thoracic rotation is required during the pull phases of freestyle and backstroke when swimming and forms a crucial component of the kinetic chain during stroke initiation as the athlete attempts to reach a position of high humeral elevation (*Blanch, 2004*). Finally in SUP, efficient paddling occurs when the trunk is rotating to transfer force through the kinetic chain and propel the paddler past the blade (*Schram, 2015*).

Limitations to range of motion (ROM) within the thoracic spine can create compensatory mechanisms that require a greater motion requirement from the scapulothoracic and glenohumeral joints leading to muscle and strength asymmetries, and increased injury risk (*Blanch, 2004*; *Spurrier, 2015*). Given the importance of thoracic spine mobility to this cohort, an assessment of rotation would be a logical inclusion in any musculoskeletal assessment of water-based athletes. The purpose of such an assessment is to develop a profile of the athlete relating to their past and current injuries and identify limitations to joint ROM. This would allow the therapist to select interventions specific to the athlete's musculoskeletal deficits which reduces injury risk and enhances athletic performance (*Furness, 2015*; *Spurrier, 2015*). Common musculoskeletal pathologies involving the cervical spine, shoulder and lumbar spine have been treated successfully by identifying inconsistencies in strength and or ROM of the thoracic spine (*Iveson et al., 2010*). The ability to quantify these thoracic asymmetries through reliable and feasible methods are limited in the literature making it difficult to comprehensively assess the thoracic spine (*Iveson et al., 2010*).

In current clinical practice, a goniometer is commonly used by physiotherapists due to its ease of use, cost effectiveness, and reliability, generally ranging from good to excellent

depending on the extremity being assessed (*Norkin & White, 2016*). Studies of reliability of the measurement of spinal movements using goniometry are limited, however, with such measures reported to have lower reliability than those of the extremities (*Burdett, Brown & Fall, 1986*; *Gajdosik & Bohannon, 1987*; *Nitschke et al., 1999*; *Youdas, Carey & Garrett, 1991*). This may be due to the complexity of the spine and position of surrounding tissues which can make it difficult to palpate anatomical landmarks to use as a reference points for goniometer positioning. These characteristics of the spine also can lead to movement of the axis or stationary arm of the goniometer, predisposing it to error. Both the individuals effort in active ROM and variations in manual force applied by the examiner in passive ROM can also be a source error in goniometry (*Norkin & White, 2016*).

A novel solution to negating the potential errors inherent to goniometry is in the use of a smart phone application. More recently, smartphones have been used to quantify joint movement and may offer equal or greater reliability than goniometry while avoiding several of the sources of error. Smartphone software applications (apps) programmed to measure joint ROM may provide health practitioners a novel clinical tool to examine joint function, detect joint asymmetry and evaluate treatment efficacy as an objective outcome measure. They are easy to use, cheap and highly accessible, with an estimated 79% of Australians under 55-year-olds owning a smart phone (*Drumm & Swiegers, 2015*). A recent systematic review evaluating mobile phone-based tools to assess joint ROM, functional assessment and rehabilitation of proprioception identified 22 articles which measured either reliability and or validity (*Mourcou et al., 2015*). Several of these articles used clinical tools as the gold standard reference (goniometer, inclinometer, scoliometer etc.) (*Jenny, Bureggah & Diesinger, 2016*; *Jones et al., 2014*; *Tousignant-Laflamme et al., 2013*); however, no study was identified that utilized a mobile phone application to determine thoracic spine rotation ROM (reliability or Validity).

Given the importance of thoracic rotation in water-based sports, simple clinical measures to quantify rotation need to be developed. Despite being used successfully in various other joints in the body, smartphone app assessment of thoracic spine rotation is yet to be investigated. Therefore, the aim of this study is to determine the reliability (intra and inter rater) and validity of the iPhone® app (Compass) when assessing thoracic spine rotation ROM in healthy individuals.

## METHODS

### Study design
Observational design assessing the reliability and concurrent validity of the iPhone® app (Compass).

### Participants
A total of thirty participants were recruited from a student University population. Sample sizes of at least 15–20 is considered adequate for reliability studies which collect continuous data (*Lexell & Downham, 2005*). Participants were excluded if they were currently experiencing back or trunk pain, had any back injury within six weeks prior to testing, had a history of spinal surgery, were younger than 18 years of age, or refused to

give informed consent to perform thoracic rotation from a seated position. The study was approved by the University Human Research Ethics committee (RO 1610) and informed consent was gained from all participants via a consent form.

## Examiners

Measurements were conducted by two final year post graduate physiotherapy students. Both students underwent several familiarization sessions using the UG and the Compass app. These sessions were supervised by two senior physiotherapists with over five years of experience in assessment of orthopedic conditions. In addition, each examiner familiarized themselves with both the Compass app and the measurement of thoracic rotation prior to any measurements. They were blinded to each other's results by independently recording data into separate spreadsheets.

## Instrumentation

For this study, a UG and an iPhone® model 6S (iPhone® is a trademark of Apple Inc, Cupertino, CA, USA) with the Compass app (Apple Pty Limited, United States) were utilised. The UG was a plastic twelve-inch goniometer (SAEHAN Grip™ Rulong 20 cm, Belgium) with moveable arms and a numerical face. The Compass app (Apple Pty Limited) comes pre-installed on all iPhones and uses the device's inbuilt magnetometer, accelerometer and GPS receiver to detect the Earth's magnetic field and orients the device to magnetic north while the device rotates through the transverse plane (*Dixon-Warren, 2012*; *Wilson & Fenlon, 2007*). It has been validated for recording ROM measurements in the transverse plane in the cervical spine (*Tousignant-Laflamme et al., 2013*). Both examiners used an Apple iPhone® 6S (64GB, model no. NKQR2ZP/A, Cupertino, CA, USA) running iOS 10.2.

## Experimental procedures

For this study thoracic rotation was defined as the combined gross axial rotation occurring throughout all thoracic spinal segments. As the shoulders, cervical and lumbar spine and hips all contribute to rotation, several minimization techniques were implemented as discussed below. There are several techniques reported in the literature for measuring thoracic rotation; however, the 'seated rotation with bar in front' technique has been reported to produce the greatest within-day inter-rater reliability (ICC: 0.87) and was therefore utilized for this study (*Johnson et al., 2012*; *Norkin & White, 2016*). This technique allows easier visualization of anatomical landmarks making it more suitable for novice examiners (*Johnson et al., 2012*).

The order of examiners, UG or Compass app and the direction in which the participant rotated were randomized using the randomization function in Microsoft Excel (v16.0; Redmond, WA, USA). After one examiner had completed all measurements, participants were asked to return five minutes later so that the next examiner could repeat the same protocol. This timeframe has previously been used in a reliability study assessing thoracic rotation (*Johnson et al., 2012*). The examiners obtained three active ROM measurements for left and right rotation for both the Compass app and the Goniometer. This provided measures for within-day intra and inter-rater reliability. All measured angles were recorded

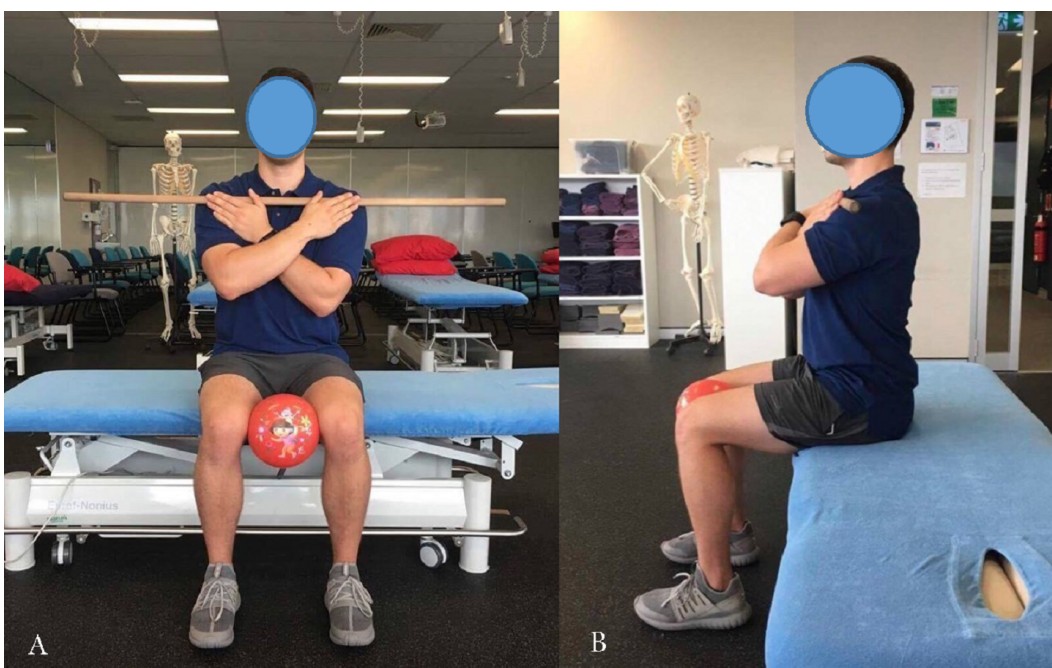

**Figure 1** Starting position for seated rotation bar-in-front.

into an Excel spreadsheet by the examiner immediately after the measurement was taken. Each examiner had their own spreadsheet to ensure true blinding between examiners.

Prior to any measurements being taken, subjects were asked to complete a series of warm up exercises to ensure task familiarity and to minimise the chance of a warmup effect on ROM or injury. This warm up involved repeated submaximal ROM thoracic rotations to the left and right five times whilst seated with their arms across their chest as per the ACSM guidelines for movement specificity (*Nadalen, 2016*). To minimize variations in participant positioning, participants were instructed to place their feet flat on the floor, knees and hips in 90° of flexion and to maintain a neutral spine position where the lumbar lordosis was maintained. To minimize contribution of movement in the hips, an inflatable ball was placed between the participant's knees. To minimize contribution of movement in the upper limbs primarily scapula-thoracic region, participants crossed their arms over their chest while holding a wooden bar which was placed at the acromioclavicular level (Fig. 1) (*Johnson et al., 2012*). The examiners provided standardized verbal (see Appendix) and manual cues to correct any deviations from this positioning throughout testing to ensure consistency of the movement performed with each trial.

## Universal goniometer measurement

The UG was positioned with the axis fixed in the transverse plane at the level of T1-T2. As previously used by *Lewis & Valentine (2010)*, T1-T2 was identified by palpating inferiorly from C7 and by palpating the spine of the scapula and moving medially to the spinous process and moving up 1–2 levels. The stationary arm of the UG was pointed to the

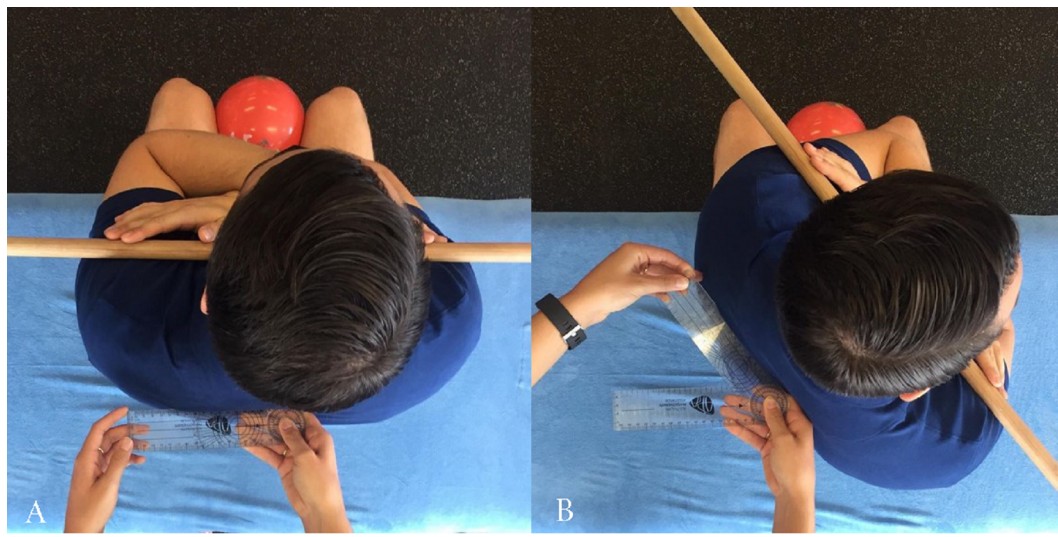

**Figure 2** **Measurement of thoracic rotation with the universal goniometer.**

opposite side of rotation and remained in the starting position while the mobile arm remained parallel to the floor, with the spine of the scapula used a reference point during rotation (see Fig. 2). Participants were asked to look straight ahead to minimise any cervical rotation. The maximum angle the participant reached at the end of their ROM was both measured and recorded by the same examiner. At the completion of measurement, the participant was instructed to return to the start position where the UG was removed from the subject and repositioned before the next measurement. Participants were allowed to rest while the examiner entered the measurement into a spreadsheet.

## Compass app measurement

To measure thoracic rotation with the Compass app, the participant was placed in the position shown in Fig. 1. The examiner placed the iPhone® approximately at the level of T1-T2 of each participant using the same palpation method described previously. The iPhone® was positioned so that magnetic north was facing directly towards the participant which was determined by the Compass app dial reading 0°. The examiner held the iPhone® firmly against the participants back at the T1-T2 level whilst the participant performed active thoracic rotation (Fig. 3) to their end ROM. The examiner followed the participant's movement by maintaining firm pressure of the iPhone® against the T1-T2 level during rotation. The participant was asked to return to the starting position which was determined by the Compass app displaying a value of 0° position before the subsequent measurement; this was actively completed by the participant. Participants were allowed a period of rest while the examiner entered each measure into a spreadsheet.

## Statistical analysis

Statistical analyses were performed using SPSS (Version 24.0; IBM Corp, Armonk, NY). Intra-rater and inter-rater reliability were determined using an intraclass correlation

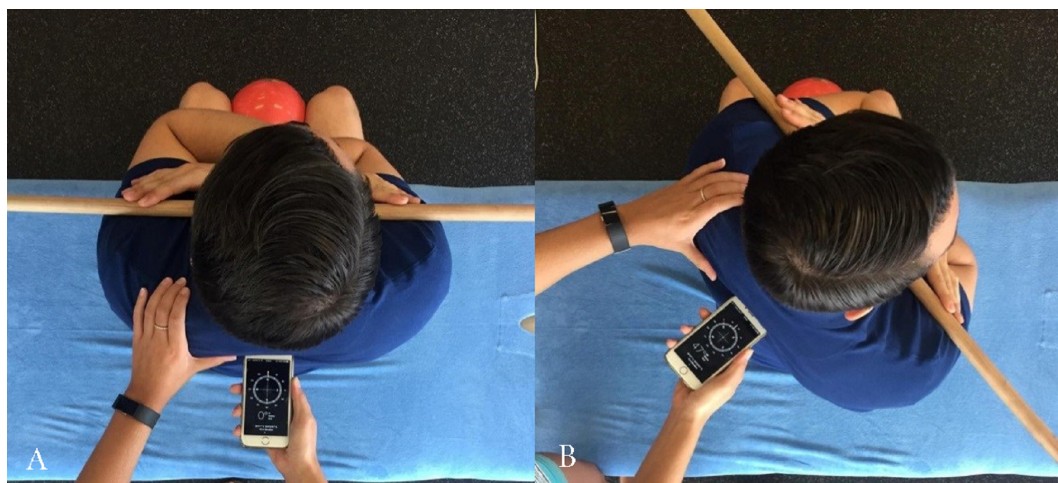

**Figure 3  Measurement of thoracic rotation with the iPhone Compass App.**

coefficient ($ICC_{3.2}$ and $ICC_{2.2}$ respectively) and associated 95% confidence intervals (CIs) (*Trevethan, 2017*). *Shrout & Fleiss (1979)* suggest that ICC values greater than 0.75 indicate excellent reproducibility, values between 0.40 and 0.75 represent fair to good reliability and values less than 0.40 indicate poor reproducibility.

To assess measurement variability, the standard error of measurement (SEM) was calculated. This measurement is defined by the equation $SEM = \sqrt{WMS}$ whereby WMS represents the mean square error term from the analysis of variance (*Lexell & Downham, 2005*). Furthermore, the SEM provides an 'absolute index of reliability' or 'typical error' associated with a measurement (*Trevethan, 2017*). The smallest amount of change that can be detected by a measure that corresponds to a noticeable change (clinically important changes) was calculated using the smallest real difference (SRD) equation $SRD = 1.96 \times SEM \times \sqrt{2}$.

To determine the construct validity of the Compass app, a Pearson's correlation coefficient was used. The linear relationship was presented graphically in a scatter plot with the associated R squared value. The level of agreement between the two devices was also presented through Bland–Altman plots with the associated 95% limits of agreement. In this, the formula: mean difference between measures $\pm 1.96 \times SD$, was used (*Bland & Altman, 2010*).

## RESULTS

### Demographics

In total, 30 participants were assessed for their thoracic rotation (20 females, 10 males, 29.8 ± 8.9 years, height 167.8 ± 8.9 cm, mass 67.9 ± 10.1 kg. Average range of motion measurements are displayed in Table 1. Measurement of thoracic rotation with the UG resulted in an average score of 63.0° (±11.3) while the Compass app averaged 60.1° (±10.7). Paired sample t-tests revealed no differences between left and right rotation for both the Goniometer and Compass app ($p = 0.279$ and $p = 0.791$ respectively).

**Table 1  Descriptive data for thoracic rotation.**

|  | $n = 30$ | $\bar{x}$ (°) | SD |
|---|---|---|---|
| Left | Goniometer | 63.5 | 11.4 |
|  | Compass app | 60.3 | 11.7 |
| Right | Goniometer | 62.5 | 12.2 |
|  | Compass app | 59.9 | 11.6 |
| Average | Goniometer | 63.0 | 11.3 |
|  | Compass app | 60.1 | 10.7 |

**Notes.**
$\bar{x}$, mean value; SD, standard deviation; $n$, number of subjects.

**Table 2  Intra-rater reliability for examiner 1 and 2 comparing the UG and the iPhone® Compass application.**

| Rater #1 (three measures each side) | | | | | | |
|---|---|---|---|---|---|---|
|  | Right | | | Left | | |
|  | ICC (95% CI) | SEM (°) | SRD (°) | ICC (95% CI) | SEM (°) | SRD (°) |
| Goniometer | 0.97 (0.95, 0.98) | 2.95 | 11.56 | 0.98 (0.97, 0.99) | 2.23 | 8.74 |
| Compass | 0.97 (0.96, 0.99) | 2.90 | 11.36 | 0.98 (0.97, 0.99) | 2.44 | 9.56 |

| Rater #2 (three measure each side) | | | | | | |
|---|---|---|---|---|---|---|
|  | Right | | | Left | | |
|  | ICC (95% CI) | SEM (°) | SRD (°) | ICC (95% CI) | SEM (°) | SRD (°) |
| Goniometer | 0.97 (0.94, 0.98) | 3.87 | 15.17 | 0.94 (0.88, 0.97) | 4.36 | 17.09 |
| Compass | 0.96 (0.93, 0.98) | 3.79 | 14.73 | 0.97 (0.95, 0.98) | 2.98 | 11.68 |

**Notes.**
ICC, Intraclass correlation coefficient; CI, confidence interval; SEM, standard error of measurement; SRD, standard real difference.

## Reliability analysis
### Intra-rater reliability
Examiners 1 and 2 had consistently high ICC values for both the UG and the Compass app as shown in Table 2. ICC values for examiner 1 were between 0.97–0.98 and between 0.94–0.97 for examiner 2 for both instruments indicating excellent reproducibility.

### Inter-rater reliability
ICC values for the UG and the Compass app for both examiners were in the good-excellent range (0.72–0.89) (Table 3). The Compass app showed slightly higher ICC values than the UG, with less variability as indicated by the upper and lower confidence intervals.

## Concurrent validity
Pearson's correlation coefficient revealed a strong and significant association ($r = 0.835$, $P < 0.001$) between the UG and the Compass app measurements. As no significant differences were seen between left and right rotation for either device an average of both movements was used in the analysis. Figure 4 presents this positive linear association between the two devices. Linear regression analysis was performed which resulted in a
**Table 3** Inter-rater reliability for the UG and the iPhone® Compass application.

| | Right (average of three measures) | | | Left (average of three measures) | | |
|---|---|---|---|---|---|---|
| | ICC (95% CI) | SEM (°) | SRD (°) | ICC (95% CI) | SEM (°) | SRD (°) |
| Goniometer | 0.85 (0.57, 0.94) | 6.33 | 17.46 | 0.72 (0.21, 0.88) | 7.85 | 21.76 |
| Compass | 0.87 (0.74, 0.91) | 5.46 | 15.13 | 0.89 (0.77, 0.95) | 5.17 | 14.33 |

**Notes.**
ICC, Intraclass correlation coefficient; CI, confidence interval; SEM, standard error of measurement; SRD, standard real difference.

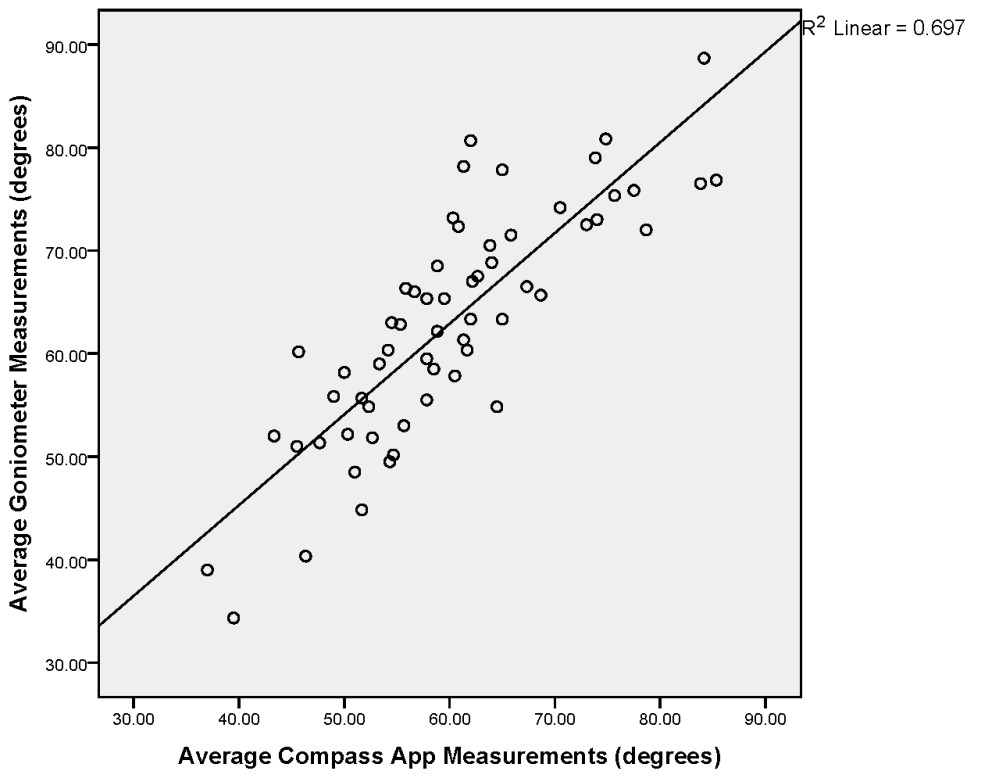

**Figure 4** Scatterplot Showing Relationship between iPhone® Compass app and Goniometer for thoracic rotation ROM.

value of $r^2 = 0.697$, indicating that approximately 70% of the variation in values obtained by the compass app can be explained by the variation of values obtained by the UG.

A Bland–Altman plot was produced to graphically represent the level of agreement between the two devices for thoracic rotation (left and right averaged) (Fig. 5). The mean difference between measures was 2.8° (SD 6.3°) and the upper and lower limits of agreements were 15.3° and –9.5°, respectively. Figure 5 graphically demonstrates the majority of data points close to the mean difference and within the 95% limits of agreement; however, values can vary by 24.8°. Systematic bias was revealed through a one-sample t test (2.8°, $P < 0.001$), indicating UG values were consistently higher when compared with the Compass app (the null hypothesis would result in a mean difference of 0°).

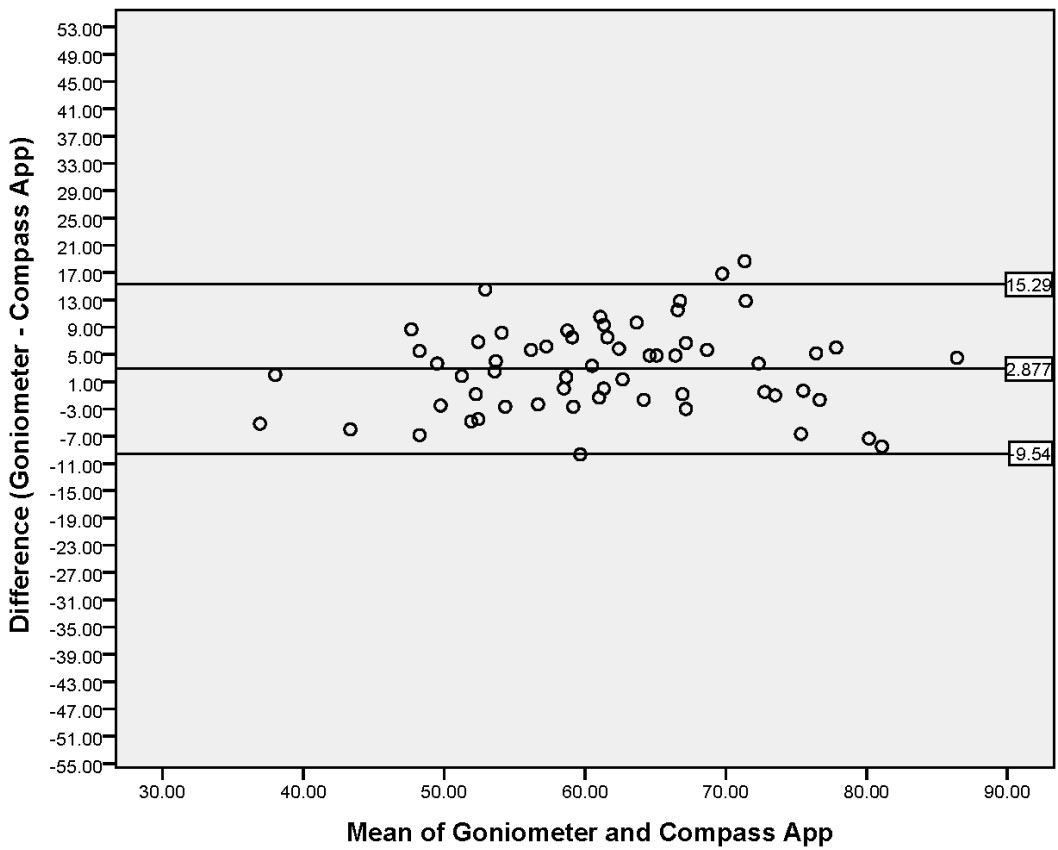

**Figure 5**  Bland–Altman Plot indicating mean difference and 95% limits of agreement between measurements from the iPhone® Compass app and Goniometer for thoracic rotation (°).

## DISCUSSION

The aim of this study was to determine the reliability (intra and inter rater) and validity of the iPhone® app (Compass) when assessing thoracic spine rotation ROM in healthy individuals. The primary findings of this study reveal that both the UG and Compass app measurements demonstrated excellent reliability; however, ICC values for the Compass app were greater for both intra-rater and inter-rater reliability. Finally, a strong and significant correlation was also shown between the devices.

The mean values for thoracic rotation ROM were 60.1° (±10.7) with the Compass app tended to be approximately 2–3° higher at 63.0° (±11.3) with the UG (Table 1). These values were similar to previous studies which report average thoracic rotation ROM to be 55.4° (±9.2) when using the same technique (*Johnson et al., 2012*). The consistently higher values recorded by the UG are thought to be related to its positioning during measurement. The moveable arm is held against the posterior surface of the shoulder and is therefore exposed to the movement of the scapula which often tilts anteriorly as participants reach their end ROM. It is hypothesized that due to the iPhone's position being localised to the T1-T2 level, its movement is not influenced by the contribution of

the scapulothoracic joints during spine rotation. Considering the above the authors believe the iPhone application provides a rotation recording which is more localised to spinal ROM than that of the UG recording'.

## Intra-rater reliability

An excellent intra-rater reliability was found when either the Compass app (ICC = 0.96–0.98; 95% CI [0.93–0.99]) or UG (ICC = 0.94–0.98; 95% CI [0.88–0.99]) was used to assess thoracic rotation ROM. While both instruments demonstrated excellent intra-rater reliability, the ICC values are marginally higher for the Compass app. Moreover, 95% CI ranges were shown to be wider for the UG compared to the Compass app, indicating greater variability when the UG is used to measure thoracic rotation. In agreement with the present findings, *Johnson et al. (2012)* reported excellent within-day intra-rater reliability for UG measurement of thoracic rotation using the same technique; however, ICC values were slightly lower and 95% CI ranges were wider (ICC = 0.87–0.91; 95% CI [0.76–0.95]). While these differences are only small, they could be explained by the differences between methodologies. The measurement technique in the current study used verbal cues with each measurement such as "rotate as far to the left/right as you comfortably can". The purpose of this was to reduce variations in the individual's effort during active ROM which is known to be a common source of error in goniometry (*Norkin & White, 2016*). *Johnson et al. (2012)* did not report use of verbal cues in their study which may have resulted in lower ICC values and wider 95% CI ranges. *Tousignant-Laflamme et al. (2013)* also used the Apple iPhone® app *Compass* to measure transverse plane joint ROM, but in the cervical spine. In contrast to our findings, the authors reported moderate to good intra-rater reliability (ICC = 0.66–0.74; 95% CI [0.39–0.87]) for the Compass app when measuring cervical rotation ROM. Their lower ICC values might be explained by the position of the iPhone® in relation to surrounding structures. In the current study, the iPhone® was placed firmly against the participants back at the T1-T2 level, as opposed to placing the compass on the participant's head in order to measure cervical ROM. This could expose the compass to additional movements outside of rotation about the transverse axis.

## Inter-rater reliability

The inter-rater reliability of the present study was excellent when using the Compass app (ICC = 0.87–0.89; 95% CI [0.74–0.95]) and good-excellent when using the UG (0.72–0.85; 95% CI [0.21–0.94]). The 95% CI ranges were shown to be wider when the UG was used compared to the Compass app. Again, this indicates greater variability of data when the UG is used to measure thoracic rotation ROM. In comparison to our intra-rater reliability findings, both devices had lower ICC values for inter-rater reliability which is a common theme among studies of joint measurement methods (*Johnson et al., 2012*; *Norkin & White, 2016*). Our ICC values for inter-rater reliability tend to align with those reported in several other studies on smartphone applications and joint ROM measurement. *Otter et al. (2015)* measured first metatarsalphalangeal joint dorsiflexion with a smartphone goniometer application (Dr. Goniometer) and a UG and reported good inter-rater reliability for the smartphone application (ICC 0.70; 95% CI [0.60–0.80]) and moderate-good inter-rater

reliability for the UG (ICC 0.69; 95% CI [0.58–0.79]). Similarly, *Mitchell et al. (2014)* reported greater ICC values for inter-rater reliability when comparing two smartphone applications (Dr. Goniometer and GetMyROM) (ICC 0.92–0.94; 95% CI [0.85–0.98]) with a UG (ICC 0.91; 95% CI [0.64–0.97]) on active shoulder external rotation ROM of 94 females. Similar to our study, the CI's reported by *Mitchell et al. (2014)* were narrower for smartphone applications than UG which again highlights the variability of data associated with UG when compared to the Compass app.

## Concurrent validity

A strong, significant correlation was found between the Compass app and UG ($r = 0.835$, $P < 0.001$), demonstrating that measures taken by the Compass app were concurrently valid when compared to the UG. In addition, linear regression analysis revealed a value of $r^2 = 0.697$, indicating that approximately 70% of the variation in values obtained by the compass app can be explained by the variation of values obtained by the UG. There was a tendency for the UG to consistently measure higher than the Compass app ($2.8°$, $P < 0.001$, Fig. 5), revealing a level of systematic bias between the two measurement approaches.

Level of agreement between the devices was explored through Bland–Altman plots and calculating the Limits of agreement (LoA). This resulted in a value of $24.8°$ (LoA $-9.5°$, $+15.3°$) highlighting that 95% of differences between measurements by the Compass app and UG will lie within a range of $24.8°$.

This form of analysis highlights that a high correlation does not mean that the two methods agree (*Bland & Altman, 2010*). Whether the difference between the measures is acceptable comes down to clinical judgement (*Bland & Altman, 2010*). It was decided by the research team that this difference was clinically unacceptable, especially if the clinician wished to use these two devices interchangeably. It was deemed clinically unacceptable as average thoracic ROM tends to be approximately 40–50 degrees (*Iveson et al., 2010*; *Johnson et al., 2012*). A difference of 24 degrees would be approximately half the normal thoracic rotation ROM and therefore using either device interchangeably would reveal significant inconsistencies. The authors recommend consistent choice of measurement instrument which will ultimately improve clinician confidence that any changes to joint ROM are a result of treatment rather than the device used.

Considering that this is the first study to assess reliability and validity of the Compass app when measuring Thoracic rotation, it is difficult to draw similarities between our results and previous research. *Tousignant-Laflamme et al. (2013)* used the Compass app to measure cervical rotation but compared values to the Cervical Range of Motion Device (CROM) as their accepted gold standard. They reported moderate validity for right rotation (ICC 0.55; 95% CI [0.23–0.76]) and poor validity for left rotation (ICC 0.43; 95% CI [0.08–0.69]). In addition, Pearson correlation values reflected lower validity when compared to our findings (R rotation: $r = 0.58$, $P < 0.01$; L rotation: $r = 0.38$, $P = 0.04$). The authors concluded that as lower results for validity were only associated with rotation, the Compass app is sensitive to electromagnetic fields in this plane of movement. In the lumbar spine, the smartphone apps TiltMeter© and iHandy© level have both demonstrated high validity ($r \geq 0.86$) when compared to a gravity-based inclinometer for measurement

of various standing lumbar spine movements and standing lumbar lordosis (*Kolber et al., 2012*; *Salamh & Kolber, 2014*). Beyond the spine, the validity of smartphone apps to measure ROM of the joints of the extremities tends to be slightly higher than the results of the present study. *Shin et al. (2012)* used a smartphone app (Clinometer) to measure various shoulder movements, and when compared to UG measurements, Pearson r values reflected a strong positive correlation between the devices ($r = 0.79 - 0.97$). *Mitchell et al. (2014)* examined the smartphone apps GetMyROM, an inclinometry-based app, and Dr. Goniometer, a photo-based app, for assessment of active shoulder ER ROM. The authors reported excellent validity when compared to measures obtained from a UG (GetMyRom: ICC 0.94; 95% CI [0.92–0.96]; Dr. Goniometer: ICC 0.93; 95% CI [0.42–0.98]). Several other studies have investigated smartphone app measurement of knee joint flexion and have found large positive correlations with UG measures. Both *Ockendon & Gilbert (2012)* and *Jones et al. (2014)* reported high Pearson $r$ values ($r \geq 0.95$) for the knee goniometer app (KGA) and Simple Goniometer app, respectively. *Milanese et al. (2014)* also used the KGA app and showed excellent validity when comparing measurements obtained from a UG for knee flexion angle which was reported as concordance correlation coefficients (CCC) (CCC 0.99; 95% CI [0.98–0.99]).

When comparing the validity data of studies which measured the joints of the extremities with our own results, it is important to consider that the validity and reliability of measurement of joint angles can be adversely affected by the complexity of the joint itself (*Gajdosik & Bohannon, 1987*). The knee, for example, is a uniplanar hinge joint while the thoracic spine is a complex multiplanar, multifaceted joint. In addition, the thoracic spine has multiple adjacent joints and its motion requires contraction of multi-joint musculature. It has been proposed that such characteristics are likely to reduce the reliability and hence validity for measurements of these joints which provides an explanation for why our validity data, while demonstrating a strong positive correlation ($r = 0.835$, $P < 0.001$), was slightly lower in comparison to the majority of studies reviewed (*Jones et al., 2014*, *Milanese et al., 2014*; *Mitchell et al., 2014*; *Norkin & White, 2016*; *Shin et al., 2012*).

## Strengths & Limitations

To authors knowledge the UG has not been validated against a gold standard such as X-ray and Computer Tomography with respect to measuring thoracic rotation ROM even though these methodologies exist (*Lam et al., 2008*). Given the fact that there appears to be several clinical methods to measure thoracic rotation (tape measure, inclinometer and goniometer) there is no clear consensus as to which tool is superior clinically (*Iveson et al., 2010*; *Johnson et al., 2012*; *O'Gorman & Jull, 1987*). Goniometers are the most commonly and easily used tools designed to measure ROM in clinical settings and therefore were selected as the comparative tool to assess the concurrent validity of the Compass app.

The measurement of thoracic rotation with an UG is inherently difficult due to the requirement of the stationary arm to remain immobile. There is a tendency for the stationary arm to move a few degrees while manipulating the moving arm. The requirement of the measurer to remain vigilant in observing the stationary arm meant that the measurer

couldn't be blinded to their own measurement. This inability to blind the assessor may lead to a degree of measurement bias. The assessors were blinded to each other's measurements, however, minimizing recall bias and improving internal validity. The Compass app is also not without limitations, which may include: measurements with an iPhone may be subject to radiofrequency noise; secondly the user is unable to calibrate the sensors within the iPhone as this is done at a manufacturing level. This may be more relevant when using older models of the iPhone. To negate this issue, it is advisable to conduct validity/accuracy assessment prior to the commencement of testing by either using a newer model of the iPhone concurrently or by using a geometric tool as a cross reference when measuring angles.

The absence of between day measurements may also be viewed as a limitation. It should be noted that clinicians will often perform measurements, treat and then reassess the effects of their treatments however over the course of a single treatment highlighting that the same day reliability is still of clinical value. Future research could investigate the between-day reliability of the Compass app to measure movements in the transverse plane since clinicians will also measure over the course of a few days or weeks during a treatment period.

The results of this study are only applicable to the healthy population which was assessed in this study and may not be applicable to people in pathological populations. Future studies should investigate the reliability of the Compass app to measure thoracic spine rotation in participants with thoracic spine injuries or pathologies that are likely to induce limitations to ROM.

Strengths of this study include the randomization of the movement direction, examiner and the device. This negated the effect of a warm up and helped to eliminate selection bias thus promoting the efficacy of the protocol (*Suresh, 2011*). The standardization of procedures including instructions to participants, the same plinth, wooden bar, ball, UG and iPhone® was also used for each participant. The authors recommend the same standardization of protocols to be used clinically if this measurement technique is applied in clinical practice.

## CONCLUSION

Measurement of thoracic spine rotation ROM has been problematic for physiotherapists due to a lack of convenient and reliable measurement techniques. Both the UG and Compass app for iPhone® offer reliable methods for measurement when the 'seated rotation bar in front' technique is adhered to, however, the intra-rater and especially inter-rater reliability of the iPhone® app (Compass) appears to be superior to UG. Clinicians may find the iPhone® app (Compass) offers greater convenience and efficiency than the UG, meaning that it could be introduced into practice with confidence that it provides reliable measurements both within and between raters. Considering the levels of agreement are clinically unacceptable the devices should not be used interchangeably for initial and follow up measurements.

## APPENDIX

(1) "Keep your feet flat on the floor, hold the ball in between your legs firm enough that it won't drop."

(2) "Cross your arms over your chest, hold the bar against your collar bones with your fingertips and maintain the pressure of the bar on your chest as you perform the movement."

(3) "Rotate to the left/right as far as you comfortably can while keeping your head in line with your shoulders."

### Funding
The authors received no funding for this work.

### Competing Interests
Justin Keogh is an Academic Editor for PeerJ.

### Author Contributions
- James Furness conceived and designed the experiments, analyzed the data, contributed reagents/materials/analysis tools, prepared figures and/or tables, authored or reviewed drafts of the paper, approved the final draft.
- Ben Schram and Justin Keogh conceived and designed the experiments, contributed reagents/materials/analysis tools, prepared figures and/or tables, authored or reviewed drafts of the paper, approved the final draft.
- Alistair J. Cox and Sarah L. Anderson conceived and designed the experiments, performed the experiments, analyzed the data, contributed reagents/materials/analysis tools, prepared figures and/or tables, authored or reviewed drafts of the paper, approved the final draft.

### Human Ethics
The following information was supplied relating to ethical approvals (i.e., approving body and any reference numbers):

Bond University Human Research Ethics Committee granted ethical approval (RO 1610).

### Data Availability
All raw data is stored in Supplemental Information.

### Supplemental Information
Supplemental information for this article can be found online at http://dx.doi.org/10.7717/peerj.4431#supplemental-information.

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
