# Peer review of "Reliability and concurrent validity of the iPhone® Compass application to measure thoracic rotation range of motion (ROM) in healthy participants"

_PeerJ, doi:10.7717/peerj.4431_

## Round 0.1 · original submission · Minor Revisions

Thank you for your submission of a well written, organized, and thoroughly analysed manuscript. Two reviewers have presented some key comments and suggestions about which you should address in detail. In particular, be sure to include an xlsx or csv version of your raw data for those who don't use SPSS. Also, please address carefully the comments by reviewer 1 on concurrent validity.

I will also add that there is a discrepancy between the results and the discussions presented in the manuscript and the abstract in regards to concurrent validity. You've stated that the high correlation between the two tools indicates good concurrent validity (abstract), but then state the Bland-Altman LOA/bias is questionable and deemed unacceptable clinically (of which I agree). I do think this puts your conclusion of good concurrent validity in question. Please comment and/or address in your resubmission.

Sincerely,
Scotty

Reviewer 1 ·

Basic reporting

No Comment.

Experimental design

No Comment.

Validity of the findings

No Comment.

Additional comments

Overall, the manuscript meets all three of the standards for publication with PeerJ, with only a few minor modifications recommended:

1. The concurrent validity portion of the experiment would be strengthened if the authors could provide evidence for the validity of the UG for measuring thoracic rotation. Each of the references provided in the introduction and methods referred specifically to the reliability of the UG. Is there some gold standard to which the UG has been compared previously for measuring thoracic rotation? Or, perhaps the UG is the current gold standard? Without this type of reference it is difficult to interpret the relationship between the two measurement techniques (comparing iPhone app with another technique that itself has not been validated). However, this may not be possible if no previous data exists. If this is the case, the authors should mention this in their discussion.
2. In table 2 it appears that SEM was greater for rotations to the right side vs left side for 3 of the 4 combinations of rater/measurement technique. Can the authors explore whether there are differences in reliability to one side vs the other for any of these combinations, and if so, briefly discuss why this might be the case. It appears that Rater #1 may have been more consistent toward one side. However, when measures from both observers are combined it appears that these differences are much smaller (table 3).
3. The authors could briefly discuss the unique challenges with using the iPhone over an analogue device like the UG. For example: 1) measurements with an iPhone could be subject to RF noise, 2) an iPhone may not be usable in certain locations (if GPS is required but connection not possible in certain buildings), and 3) one cannot calibrate sensors in the iPhone (they are set at the factory and may lose calibration over time) – an older iPhone may therefore be less accurate.

Minor corrections:
Methods:
Line 122: …sessions “were”
Line 128: …Compass app (Apple…) “were” utilised.
Line 131: Does the app also make use of the phone’s gyroscope?
Lines 138-143: Can the authors define thoracic rotation from an anatomical perspective for a broader scientific audience? For example, does motion occur exclusively between thoracic vertebrae? Can motion occur between other articulations?
Line 196: …reliability “were” determined…



Discussion:
Lines 267-269: Would the authors therefore speculate that the iPhone app was more accurate? Perhaps this should be stated? This might be a good place to address the validity question of the UG (mentioned above).
Line 286: …transverse “plane” joint ROM.
Lines 288-293: Can the app be used with other axes of iPhone rotation? Perhaps using different axes of the iPhone may influence accuracy? The authors briefly mention this in lines 339-343.
Lines 316-318: Can the authors re-consider their definition for r2 (coefficient of determination)? The classic definition is that it describes the amount of variance in one measure that can be explained by the variance in a second measure.
Lines 327-329: Can the authors provide a little more rationale for their decision? Is there a criterion level of difference that would be acceptable in a clinic?
Line 333: …measuring “t”horacic rotation…

Figures:
All: consider including information about number of subjects/number of measures included in each table/figure (i.e. n=?)
Figure 3 – Consider increasing the font of r2 data and best-fit line equation, and move the equation to the side so that it doesn’t obscure data points.

Reviewer 2 ·

Basic reporting

This article is written in clear and unambiguous English.
The introduction is generally good,but it should not be mentionned an "unpublished review". and all the more that there is one which could suit : https://www.hindawi.com/journals/bmri/2015/328142/
The introduction also does not insist on articles that validate the use of internals Smartphone sensors comparing to godl standard in the clinical pratice.

The structure of the article is confomr to acceptable format. Figures and tables are relevant.

Raw data are available but only in SPSS format and with no meta-data explanation. Although it is still possible to convert the data and interpret meta-data, it could be more usefull to get raw data in csv format and with a text document which explain meta-data.

This submission is well 'self-contained'.

Experimental design

Research question is clearly and well defined. Investigation respect the state of art of such research so technical level is good.
Methods are described with sufficient detail and information to replicate, although the writing of certain formulas would make it possible to improve even more this good work.
In "instrumentation part", line 128, it would be nice to clearly define which models of inertial units used in these iPhones.
In the Compass App measurement, line 189, it is not clear if the participant return actively to starting position without help of examiner. The sentence seems ambiguous and should be reformulated for clarity.

Validity of the findings

Novelty could be juged insufficient but the limits of the work are very clearly defined and that is very appreciable. So benefit to literature is clearly state.
The discussion is of a good level and all the limits of the results are discussed clearly and well documented.
Conclusion is well limited to supporting results.

Additional comments

This work is a quality work concerning the reliability and validaty of iPhone app when assessing thoracic spine rotation in healthy individuals. It is rather well documented and very well discussed. The limits being very well defined, I encourage the authors to continue and go further in the validation of this tool.

---

## Round 0.2 · accepted · Accept

Thank you for your attention to detail and your comprehensive responses to the comments provided by the reviewers and I.

Congratulations on a great paper!